# Magnetically Tunable Liquid Crystal-Based Optical Diffraction Gratings

**DOI:** 10.3390/polym12102355

**Published:** 2020-10-14

**Authors:** Dejan Bošnjaković, Nerea Sebastián, Irena Drevenšek-Olenik

**Affiliations:** 1Faculty of Mathematics and Physics, University of Ljubljana, Jadranska 19, 1000 Ljubljana, Slovenia; Irena.Drevensek@ijs.si; 2Faculty of Electrical Engineering, Computer Science and Information Technology, Josip Juraj Strossmayer University of Osijek, Kneza Trpimira 2B, 31000 Osijek, Croatia; 3J. Stefan Institute, Jamova cesta 39, 1000 Ljubljana, Slovenia; nerea.sebastian@ijs.si

**Keywords:** ferromagnetic materials, liquid crystals, optical diffractive structures, transmission gratings

## Abstract

We present a theoretical analysis of optical diffractive properties of magnetically tunable optical transmission gratings composed of periodically assembled layers of a polymer and a ferromagnetic liquid crystal (LC). The orientational structure of the LC layers as a function of an applied magnetic field is calculated by minimization of the Landau-de Gennes free energy for ferromagnetic LCs, which is performed numerically and also analytically by using the one-constant approximation and the approximations of the high and the low magnetic fields. Optical diffractive properties of the associated diffraction structure are calculated numerically in the framework of rigorous coupled-wave analysis (RCWA). The presented methodology provides a basis for designing new types of diffractive optical element based on ferromagnetic LCs and simulating their operation governed by the in-plane magnetic field.

## 1. Introduction

Optical gratings are central components of diffractive optical elements (DOEs) that are used in various optical devices to manipulate optical beams in order to create a desired far-field intensity profile of the beam. They can affect the amplitude as well as the phase of an optical field [1,2,3,4,5,6,7,8,9]. One of their advantageous properties is tunability, which can be obtained by the application of external stimuli, such as electric [4,10,11,12,13], magnetic [14,15,16,17], and mechanical or optical fields [18,19,20].

Liquid crystals (LCs) are very suitable materials for the construction of tunable DOEs, as they are very sensitive to various external stimuli [21,22,23] and at the same time exhibit a strong optical anisotropy (birefringence). This unique combination makes them very important materials in modern photonics technologies [24]. LCs exhibit properties of liquids, since they are composed of molecules that do not possess a long-range positional order, or possess such order only in one or two spatial dimensions. This allows them to flow like liquids. Nevertheless, they exhibit also properties of solids (crystals), since they possess a long-range orientational order, giving rise to optical birefringence and other anisotropic properties that are characteristic for crystals.

A wide range of fabrication techniques has been developed to construct LC-based optical diffractive gratings. They can be divided into two main categories that are associated with surface-based and volume-based LC patterning. Surface-based patterning is typically obtained by the use of striped electrodes that generate a periodic electric field within the LC medium or by patterning of alignment layers that govern the LC orientation in contact with the surrounding surfaces [10,23,25,26]. Volume-based patterning relies on the use of mixtures of LCs with photopolymeric materials and patterning is achieved by photolithographic methods, such as conventional mask-based optical lithography, holographic lithography and direct laser writing [27,28,29,30].

Despite the fact that LC materials are very sensitive to an external electric field, i.e., the application of a voltage of ~1 V can cause optical refractive index modification of ~0.2, the associated need for metallic electrical contacts (wire connection to ITO layers) can sometimes be a disadvantage, for instance in medical applications. This problem can be avoided by using magnetic instead of electric field to control the LC orientation. However, since the anisotropy of magnetic permeability of conventional LC materials is very low (~10^−6^), a desired responsivity to the magnetic field is usually obtained by doping magnetic nanoparticles. Recent investigations have shown that doping with ferromagnetic nanoplatelets induces the formation of a ferromagnetic nematic LC phase [31,32], which enables LC reorientation with magnetic fields as low as a few mT [33]. Such LCs are very attractive for the construction of various kinds of magneto-optical device, among them also magnetically tunable DOEs.

In this work, we present a theoretical analysis of optical diffractive properties of magnetically tunable optical diffraction gratings based on ferromagnetic LCs. The orientational structure of the LC medium in a periodic polymeric scaffold as a function of an applied magnetic field is calculated by minimization of the Landau-de Gennes free energy for ferromagnetic LCs [34,35], while optical diffractive properties of the associated diffraction structure are calculated by the use of rigorous coupled-wave analysis (RCWA) [36,37,38,39]. The results obtained show good agreement with our previously reported experimental data [40,41]. Besides this, we also show that the presented methodology can be used to design new types of DOE based on ferromagnetic LCs and simulate their operation governed by the in-plane magnetic field.

## 2. Grating Assembly

As shown in our previous work, a tunable optical diffraction grating can be constructed by introducing a mixture of liquid crystalline material and ferromagnetic nanoplatelets into a periodic scaffold of parallel polymeric ribbons, where polymer ribbons are made beforehand from a negative photoresist material (SU-8 polymer materials). The periodic scaffold was fabricated by the direct laser inscription (DLW) method (Figure 1a) [42,43,44,45]. ITO-coated glass plates, that had no additional surface treatment, support the grating structure from the bottom and the top sides and the entire assembly is glued together at the edges. The main parameters that define the optical properties of the resulting grating are the grating period *Λ*, the width of the polymer ribbons *d* (*w = Λ − d* is the gap between two adjacent ribbons), and the height of the ribbons *D*. The values of *Λ* and *d* depend on the DLW procedure, while the value of *D* is determined by the thickness of the initial spin-cast polymer film.

Local orientation of LC molecules in the ferromagnetic nematic phase is described by a unit vector **n**(**r**) called the director, while local magnetic properties are described by the magnetization **M**(**r**) (Figure 1b). From the perspective of optical properties, nematic LCs are uniaxial optically birefringent materials with an optical axis parallel to **n**. When there is no magnetic field present, all LC molecules are homogeneously aligned along with the polymer ribbons, as shown in Figure 1c. The alignment is induced by the surface relief structure on the side walls of the ribbons, which is formed as a consequence of optical interference between the incident and reflected laser beams during the DLW process [42,46]. When a magnetic field **B** is applied in the plane of the grating structure (*xy*-plane), it causes an in-plane reorientation of the LC molecules towards the direction of the magnetic field **B**, as shown in Figure 1d. The reorientation process is spatially inhomogeneous, as boundary conditions at the surfaces of the ribbons try to preserve the orientation of **n** parallel to the ribbons, i.e., along the *y*-axis [47], while the magnetic field-induced torque pushes **M** and subsequently also **n** into the orientation parallel to **B**. The resulting orientational profile can be described as **n** = (sin(*θ*(*x*)), cos(*θ*(*x*)), 0), and **M** = *M*(sin(*ψ*(*x*)), cos(*ψ*(*x*)), 0) where *θ*(*x*) and *ψ*(*x*) are inclination angles of **n** and **M** with respect to the *y*-axis, respectively. This leads to a spatial modulation of the orientation of the optical axis of the medium. Consequently, for an optical beam propagating along the *z*-axis, such a system acts as magnetically tunable optical polarization grating [40].

In our previous theoretical modeling of the assembly described above, the LC medium between polymer ribbons was assumed to rotate homogeneously, i.e., independent of the coordinate *x*, [40,41], which gave only a qualitative description of the observed features. Besides this, no relation between the LC rotation angle and the applied magnetic field was taken into account. The primary steps forward in this work are that (i) we consider the inhomogeneous reorientation of the LC medium and also (ii) the relationship between such reorientation and the applied magnetic field.

## 3. Orientational Profile of the Liquid Crystalline Medium

### 3.1. Minimization of Free Energy

As mentioned, the ferromagnetic nematic phase can be described by two macroscopic variables—the director **n** and the magnetization **M**. The former is associated with the preferential orientation of the LC molecules and the latter with the preferential orientation of the magnetic nanoplatelets, where properties of magnetic nanoplatelets are explained in references [32,34]. Both are coupled through the LC molecules anchoring to the platelets surface, being such in this case that in the absence of external field **n** and **M** are parallel. The behavior of the ferromagnetic LC in an external magnetic field is described by the free energy density [34],
(1)$$f=f^{F}-\frac{1}{2}A{\left(M\cdot n\right)}^{2}-\mu_{0}M\cdot H+f_{0}$$
where *f ^F^* is the Frank elastic energy density [48], while the second term describes coupling between the director field and the magnetization, the third coupling between the magnetization and the external magnetic field and the last one free energy of the homogeneous ferromagnetic nematic phase. The last term depends on the degree of ordering of the molecules along the director **n** and on the magnitude of magnetization **M**. The latter is assumed to be constant in our case so it will not be taken into consideration, since we also assume that $\left|M\right|=M_{0}$, i.e., that variations of the magnitude of the magnetization are negligible. The parameter *A* is defined as $A=\gamma \cdot \mu_{0}$, where *γ* is the coupling constant, which determines the relative orientation of **M** with respect to **n** and *μ*_0_ is vacuum permeability.

In accordance with Figure 1b, we are interested in the situation in which B=B(sinπ/4,cosπ/4,0), where $B=\mu_{0}H$. The corresponding free energy density is,
(2)$$f=\frac{1}{4}(K_{1}+K_{3}+(K_{1}-K_{3})(\cos2\theta )){\left(\frac{\partial \theta }{\partial x}\right)}^{2}-\frac{1}{2}AM_{0}^{2}\cos^{2}(\theta -\psi )-BM_{0}\cos\left(\psi -\frac{\pi }{4}\right)$$
where constant term *f*_0_ is omitted and where *K*_1_ and *K*_3_ are the Frank elastic constants for splay and bend deformation, respectively [48,49]. In accordance with our experimental studies, which are performed with the conventional liquid crystalline mixture E7 doped with the barium hexaferrite nanoplatelets, we took in our calculations *K*_1_ = 11 pN, *K*_3_ = 14.5 pN [34,50], *M*_0_ = 200 A/m and *γ* = 70 [34]. The Equation (2) was used to calculate the orientational profile of the ferromagnetic LC between two polymer (SU-8) ribbons via numerical minimization of the free energy by using the Wolfram Mathematica. The space between two ribbons was discretized into slices with the width Δ*x* = *w*/(*N* − 1), where *N* is the number of discretization cells. We found that using *N* = 51 was sufficient, as for this *N* the thickness of the slices becomes already significantly lower than the wavelength of visible light.

In the free energy minimization process, appropriate boundary conditions have to be taken into account. These are dictated by the interfacial interaction (anchoring) between the LC molecules and the polymer ribbons. The corresponding term for the surface energy density is,
(3)$$f^{s}=-\frac{1}{2}W{\left(n_{s}\cdot n\right)}^{2}$$
where *W* is the strength of the surface anchoring and **n*_s_*** is the preferred direction of **n** at the surface. In our case $n_{s}={\hat{e}}_{y}$.

In order to solve the problem analytically, some additional approximations were necessary. We used the one-constant approximation *K* = *K*_1_ = *K*_3_ and considered only two limiting cases—the high and the low magnetic fields [51]. In the limit of high magnetic fields it was assumed that *θ*(*x*) and *ψ*(*x*) are both close to *π*/4, while in the limit of low magnetic fields it was assumed that they were both close to zero. With these approximations, in the limit of high magnetic fields, the Equation (2) becomes,
(4)$$f\approx \frac{1}{2}K{\left(\frac{\partial \theta }{\partial x}\right)}^{2}+\;\frac{1}{2}{AM}_{0}^{2}{(\theta -\psi )}^{2}\;+\;\frac{1}{2}{BM}_{0}{\left(\psi -\frac{\pi }{4}\right)}^{2}$$
where constant terms and terms of the Taylor series higher than *O*(*θ*^2^) and *O*(*ψ*^2^) are omitted.

To obtain the equilibrium solution, the free energy was minimized by using the Euler-Lagrange equations for both angles [52]. The solution for *ψ*(*x*) is:(5)$$\psi (x)=\frac{AM_{0}\theta (x)+B\frac{\pi }{4}}{AM_{0}+B}$$
and the solution for *θ*(*x*) is,
(6)$$\theta (x)=\frac{\pi }{4}+C_{1}e^{qx}+C_{2}e^{-qx}$$
where,
(7)$$q^{2}=\frac{{AM}_{0}^{2}}{K}\frac{B}{{AM}_{0}+B}$$
In order to find the values of the constants *C*_1_ and *C*_2_ we applied the boundary condition associated with the requirement that the equilibrium solution for the director **n** at surfaces must satisfy the anchoring condition, which can be described as [52]:(8)$$K\frac{\partial \theta }{\partial x}\nu +\frac{\partial f^{s}}{\partial \theta }=0$$
where *ν* is −1 at *x* = 0 and +1 at *x* = *w*. For strong anchoring (*W* >> *K*/*w*), the resulting solution for *θ*(*x*) is,
(9)$$\theta (x)=\frac{\pi }{4}-\frac{\frac{\pi }{4}}{1+q\xi \tanh(qw/2)}\frac{\cosh(q(x-w/2))}{\cosh(qw/2)}$$
while for weak anchoring (*W* << *K*/*w*), it follows:(10)$$\theta (x)=\frac{\pi }{4}-\frac{1}{2}\left(-\xi q\tanh\left(\frac{qw}{2}\right)+\sqrt{\xi^{2}q^{2}\tanh^{2}\left(\frac{qw}{2}\right)+2}\right)\frac{\cosh(q(x-w/2))}{\cosh(qw/2)}$$
where *ξ = K/W*. At this point, we would like to mention that large magnetic fields and large anchoring strengths have opposite effects. Large magnetic fields tend to rotate the director **n** in the direction of **B**, while large anchoring strengths tend to keep **n** in the preferred direction dictated by the polymer surface.

When the one-constant approximation and the limit of low magnetic fields are applied to the Equation (2), it follows:(11)$$f\approx \frac{1}{2}K{\left(\frac{\partial \theta }{\partial x}\right)}^{2}+\frac{1}{2}AM_{0}^{2}{\left(\theta -\psi \right)}^{2}-BM_{0}\frac{\sqrt{2}}{2}\left(\psi -\frac{\psi^{2}}{2}\right)$$
where constant terms and terms of the Taylor series higher than *O*(*θ*^2^) and *O*(*ψ*^2^) are omitted. The solution for *ψ*(*x*) is:(12)$$\psi (x)=\frac{AM_{0}\theta (x)+B\frac{\sqrt{2}}{2}}{AM_{0}+B\frac{\sqrt{2}}{2}}$$
and the corresponding solution for *θ*(*x*) is:(13)$$\theta (x)=1-\frac{1}{1+p\xi \tanh(pw/2)}\frac{\cosh(p(x-w/2))}{\cosh(pw/2)}$$
where,
(14)$$p^{2}=\frac{AM_{0}^{2}}{K}\frac{B\frac{\sqrt{2}}{2}}{AM_{0}+B\frac{\sqrt{2}}{2}}$$

In the limit of low magnetic fields, weak and strong anchorings do not give different results, because in both cases the Taylor series expansion of Equation (8) is performed around *θ* = 0. Low magnetic fields and strong anchorings both cause the director **n** not to rotate much from its initial orientation along the *y*-axis.

### 3.2. Comparison of Analytical and Numerical Results

The numerical and analytical results for *θ*(*x*) obtained in the limit of high magnetic fields are shown in Figure 2. Numerical results are presented by dots, while solid lines correspond to the analytical results associated with Equation (9) and dashed lines to the analytical results obtained by the Equation (10). The results are given for magnetic fields *B* = 50 mT (Figure 2a) and *B* = 10 mT (Figure 2b). In the analytical calculations we took *K* = 12.8 pN, which is the average value of *K*_1_ and *K*_3_ for the LC compound E7, while the distance between the polymer ribbons was taken to be *w* = 3.2 μm, in accordance with our experiments [40,41]. Consequently, the ratio of *K/w*, which defines the regimes of strong and weak anchoring, has a value of 4.0 µJ/m^2^. However, it should be noted that the values of *B* and *W* at which different approximations described above become valid depending also on the values of the parameters *M*_0_ and *γ.*

In Figure 2 one can notice that for the anchoring strength *W* = 10 μJ/m^2^, the analytical results obtained by Equation (9) exhibit a much better agreement with the numerical results than the analytical results obtained by Equation (10), while for the anchoring strength *W* = 1 μJ/m^2^ the situation is just the opposite. This is observed for *B* = 50 mT as well as for *B* = 10 mT; therefore, for the investigated system both fields can be considered as high magnetic fields.

Consequently, the behavior associated with the limit of low magnetic fields was investigated for *B* < 1 mT. Figure 3 shows a comparison of the numerical results with the analytical results, where it can be noticed that for *W* = 10 μJ/m^2^ (Figure 3a) the analytical results obtained by Equations (9) and (13) both exhibit a slight deviation from the numerical results, while for *W* = 1 μJ/m^2^ (Figure 3b) the agreement is better.

## 4. Calculation of Optical Diffraction Properties

In the analysis of optical diffraction properties, polymer ribbons are considered to be an optically isotropic material with the refractive index *n_p_* = 1.57 [53] and the LC medium an optically uniaxial material with the extraordinary refractive index *n_e_* = 1.73 and the ordinary refractive index *n_o_* = 1.52 [54]. Due to the very low concentration of magnetic nanoplatelets in the ferromagnetic LC mixture (*c* < 0.3 wt. %), the values of the refractive indices for the pure LC material (E7) were taken. In the absence of an external magnetic field, the optical axis of the LC medium points along the *y*-axis (see Figure 1) and the incident optical beam used to probe the diffraction properties propagates along the *z*-axis. The beam is linearly polarized either along the *y*-axis (s-polarization) or along the *x*-axis (p-polarization).

In accordance with the conventional classification, the investigated gratings belong neither to the thin (Raman–Nath) nor to the thick (Bragg) diffraction regime, but, to the so-called mixed diffraction regime [55]. For such a regime Moharam and Gaylord [36] proposed the use of rigorous coupled wave analysis (RCWA), which is based on the coupled wave theory (CWT) of propagation of optical waves in periodic media. The basic idea is to describe the optical field as a superposition of waves that are composed of the incident wave and the diffracted waves whose wave vectors are defined by the Fourier transform of the spatial dependence of the dielectric permittivity of the grating assembly. The analysis is based on Maxwell’s equations for optical grating structure, which can be for the configuration as shown in Figure 1 written as [39]:(15)$$\nabla \times E=i\omega \mu_{0}\mu_{r}H$$
(16)$$\nabla \times H=-i\omega \varepsilon_{0}\underline{\varepsilon }(x)E$$
where **E** and **H** are electric and magnetic field vectors, respectively, *ω* is angular frequency and $\underline{\varepsilon }(x)$ is dielectric tensor of the grating region. Because of the structural periodicity, **E** has a form,
(17)$$E(r,z)=\sum_{G}E_{G}(z)e^{i(k+G)r}$$
where $k=(\omega /c){\hat{e}}_{z}$ is the wave vector of the incident beam and $G=\pm m(2\pi /\Lambda ){\hat{e}}_{x}$ is the grating vector with *Λ* being the grating period and *m* an integer. Also, **H** can be written in a similar way. Searching for solutions of Equations (15) and (16) with the ansatz described above generates an infinite set of differential equations, which is truncated to a finite set of equations that can be written in the form of matrix wave equation [36,37]:(18)$$\frac{\partial^{2}}{\partial {\tilde{z}}^{2}}\left[\begin{array}{c} S_{x}\\ S_{y} \end{array}\right]-\Omega^{2}\left[\begin{array}{c} S_{x}\\ S_{y} \end{array}\right]=0$$
where ${\tilde{z}=k}_{0}z$, and **S***_x_* and **S***_y_* are column vectors containing complex amplitude coefficients of various spatial harmonics of **E** and $\Omega^{2}=PQ$, where **P** and **Q** are scattering matrices [39]. Solving the system of Equation (18) inside the grating region leads to the eigenvalues and eigenvectors for **E**. These are then used to determine the eigenvalues and eigenvectors for **H**. The diffraction efficiencies of various outgoing diffracted beams can be determined by using boundary conditions on the interface with the surrounding medium, in our case bottom and top glass plates.

The associated numerical procedure can resolve light propagation in optical grating structures with a broad range of grating periodicities and thicknesses. To perform such a calculation, we used the Stanford Stratified Structure Solver (S^4^) program, which is very convenient for the analysis of layered periodic structures via the RCWA [56].

In the aforementioned calculation, the LC material is described with its dielectric tensor:(19)$$\underline{\varepsilon }=\left(\begin{array}{ccc} \varepsilon_{⊥}+\varepsilon_{a}\sin^{2}\theta (x) & \varepsilon_{a}\sin\theta (x)\cos\theta (x) & 0\\ \varepsilon_{a}\sin\theta (x)\cos\theta (x) & \varepsilon_{⊥}+\varepsilon_{a}\sin^{2}\theta (x) & 0\\ 0 & 0 & \varepsilon_{⊥} \end{array}\right)$$
where $\varepsilon_{⊥}={\left(n_{e}\right)}^{2}$ and $\varepsilon_{a}={\left(n_{e}\right)}^{2}-{\left(n_{o}\right)}^{2}$. For calculation of the reorientation angle *θ*(*x*) as a function of the magnetic field, we used the analytical and numerical results presented in the previous section.

The obtained results for diffraction efficiencies of different diffraction orders as a function of *B* are shown in Figures 5, 6 and 8. The diffraction efficiency *η**_i_* of the *i*-th diffraction order was calculated as:(20)$$\eta_{i}=\frac{I_{i}}{I_{-2}+I_{-1}+I_{0}+I_{1}+I_{2}}$$
where *I_i_* is intensity of the selected diffraction peak and the denominator is sum of intensities of the 0th, ±1st, and ±2nd order peaks. This definition is typically used in the experimental investigations, in which intensities of the 3rd and higher-order peaks are usually negligible, to compensate for the effects of incoherent scattering and absorption.

### 4.1. Comparison to Previously Investigated Grating Structures

The parameters of the optical diffraction grating with which we started the calculation were: the thickness of the grating *D* = 9 μm, the grating period *Λ* = 5 μm, and the distance between polymer ribbons *w* = 3.2 μm. Those values were deduced from diffraction experiments with empty gratings, i.e., before the LC material was introduced into the polymeric scaffold. Then the scaffold was filled with the ferromagnetic LC. Figure 4 shows the experimentally obtained dependencies of diffraction efficiencies of the 0th, 1st and 2nd diffraction orders on applied external magnetic fields reported in our previous work [40].

Figure 5 gives the numerical calculations of the diffraction efficiencies of the 0th, 1st, and 2nd diffraction orders as a function of *B* for the anchoring strength *W* = 5 μJ/m^2^. The curves for which *θ*(*x*) were obtained by Equation (9) and are given as solid lines and those for which *θ*(*x*) was obtained by Equation (10) are given as dashed lines. In both cases, for *B* ≤ 2 mT the Equation (13) was used. The reorientation of the director **n** from its initial direction along the *y*-axis is relatively minor for all the applied magnetic fields, so diffraction properties change with the field only relatively little. This is because for strong anchoring there can exist a large difference between the directions of **n** and **M**, for high magnetic field (50 mT) angle *ψ*(*x*) ~35° and angle *θ*(*x*) ~10° so even when **M** becomes parallel to **B**, i.e., when *ψ*(*x*) ~45°, the values of *θ*(*x*) are still far below 45° [57].

If the anchoring strength decreases, field-induced reorientation of **n** becomes larger and, consequently, field-induced modifications of diffraction properties become more profound. This effect can be noticed in Figure 6, which shows the numerical calculations of diffraction efficiencies of the 0th, 1st, and 2nd diffraction orders as a function of *B* for *W* = 1 μJ/m^2^. In the insets, one can notice that the analytical results for weak anchoring obtained by the Equation (10) give better agreement with the numerical results than the analytical results for strong anchoring obtained by Equation (9). By comparing Figure 6a,b, one can see that for large values of *B* the diffraction efficiencies of the same diffraction orders for both polarizations (s and p) become quite similar. This feature resembles the observations reported in our previous experimental study (see Figure 4), therefore we conclude that the anchoring strength *W* between the polymer ribbons and the ferromagnetic LC in the experimentally investigated structures is below 1 μJ/m^2^. The anchoring strength of the pure LC (E7) in contact with SU-8 polymer ribbons was measured to be *W* = 13 μJ/m^2^ [42]. However, its nanoplatelet-doped ferromagnetic LC variant is used in the experiments with a magnetic field. We assume that adhesion of nanoplatelets on to the sidewalls of the ribbons modifies interaction between the LC and the sidewalls, so that the anchoring strength is reduced [58,59].

The deviation of numerical calculations from the experimental results is attributed to the fact that the only parameter that was varied in our simulations was the surface anchoring strength *W*, while all other parameters had fixed values as found in the literature or deduced from diffraction experiments on empty gratings. However, the effective thickness of grating *D* might change after the filling process with the LC, because LC orientation at the top and bottom surfaces might be partially out of the plane. Besides this, refractive index of the SU-8 polymer ribbons *n_p_* may differ from that reported in the literature due to different composition of different commercial prepolymer mixtures.

### 4.2. Design of New Grating Structures

One of the simplest ways to modify the diffractive properties of the above-described structures is to change the thickness *D* of the grating assembly (see Figure 1). In practice this can be achieved by changing the parameters of the spin coating process of the initial polymer film. Figure 7 shows dependencies of the 0th, 1st, and 2nd diffraction orders as a function *D* in the absence of a magnetic field. In this case, s-polarization corresponds to the extraordinary eigen-polarization and p-polarization to the ordinary eigen-polarization of the LC medium. Because refractive index mismatch between the polymer ribbons and the LC layers for the former (*n*_e_
*− n*_p_) = −0.16 is considerably larger than for the latter (*n*_o_
*− n*_p_) = 0.05, diffractive properties for s-polarization vary with *D* much more than for p-polarization.

An interesting case for further exploration is *D* = 7.75 μm. For this *D*, in zero magnetic field, the diffraction efficiency of the 1st diffraction order for s-polarization is very low and for p-polarization is quite high, so diffraction properties are highly polarization sensitive. However, as can be seen in Figure 8, when magnetic field is applied, this difference diminishes. At *B* = 50 mT, for the weakest considered anchoring (*W* = 0.5 μJ/m^2^), diffraction efficiencies for the two polarizations differ by less than 10%. Therefore, in such a grating structure a magnetic field can be used to tune the polarization selectivity of the diffractive properties.

From Figure 8 it follows that the best tunability can be obtained by utilizing very weak anchoring of the LC medium at the interface with the polymer walls. However, there exists also a drawback of this strategy, which is associated with the fact that weak LC anchoring generally leads to long response times of the orientational structure [60]. Therefore, when switching times between different diffractive states are important, a good compromise between the tunability range and the dynamic response needs to be found.

## 5. Discussion and Conclusions

In this work, we present a computational approach to theoretical modeling of the LC-based optical diffractive gratings that correspond to the so-called mixed grating regime. This regime occurs when the thickness of the grating assembly and the periodicity of the grating structure are of similar size, which quite often appears in experimental investigations. In this study, we consider that magnetic field-induced LC reorientation takes place only in the plane of the grating assembly (*xy*-plane in Figure 1), i.e., we presume fully in-plane reorientation. We improve the simulations with inhomogeneous orientation profile of the director **n**, which is determined numerically and analytically, and the diffraction efficiency was determined depending on the external magnetic field. The research was based on the anchoring energy and to see its influence as a boundary condition on the polymer–LC interface and its influence on the diffractive properties of magnetically tunable optical diffraction gratings. In experiments top and bottom walls are composed of an ITO-coated glass plates and typical anchoring strength of LCs at ITO surface is ~1 μJ/m^2^ [61,62]. However, in our simulations we considered that surface anchoring strength at top and bottom layers is zero. Adding finite anchoring strength at top and bottom layers leads to further improvement of our model, where we should take into account surface interaction with the ITO- coated glass plates on the top and bottom of the assembly, which can lead to in-plane and out-of-plane LC reorientation. Consequently, instead of two-dimensional, three-dimensional director field would need to be calculated via the free energy minimization.

Another challenge is to model the described grating assembly for the cases when, in addition to an external magnetic field, also an external electric field is applied. To the best of our knowledge such dual field-driven grating configurations have not been studied yet either theoretically or experimentally. They are expected to exhibit much shorter switching times than the standard single field-driven gratings because, for them both, switching-on and switching-off processes can be controlled by the applied fields instead of by the boundary conditions. Therefore, we believe that our approach can facilitate the design of exciting novel LC-based diffractive optical structures and simulations of different modes of their operation.

## Figures and Tables

**Figure 1 polymers-12-02355-f001:**
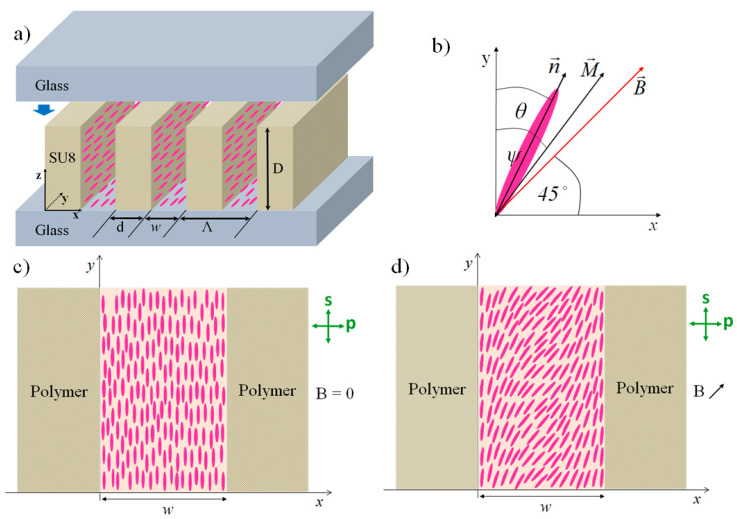
(**a**) Schematic drawing of the periodic assembly of liquid crystal (LC) channels separated by polymeric ribbons fabricated from the photoresist polymer SU-8. Stripes indicate polymer ribbons, and pink ellipsoids indicate the LC molecules. (**b**) Local orientation of the nematic director **n** and of the magnetization **M** under the influence of the external magnetic field **B** is indicated, where *θ*(*x*) and *ψ*(*x*) indicate inclination angles of **n** and **M** with respect to the *y*-axis, respectively. (**c**) Schematic drawing of the homogeneous alignment of the LC medium between two polymer ribbons at *B* = 0. Alignment along the *y*-axis is induced by the surface relief structure present on the sidewalls of the ribbons. (**d**) Schematic drawing of the in-plane reorientation of the LC molecules induced by an external magnetic field tilted at 45° with respect to the *y*-axis. Green arrow-ended lines on the right side of the drawings indicate two orthogonal linear polarizations for optical radiation propagating along the *z*-axis that are customarily denoted as s- and p-polarization.

**Figure 2 polymers-12-02355-f002:**
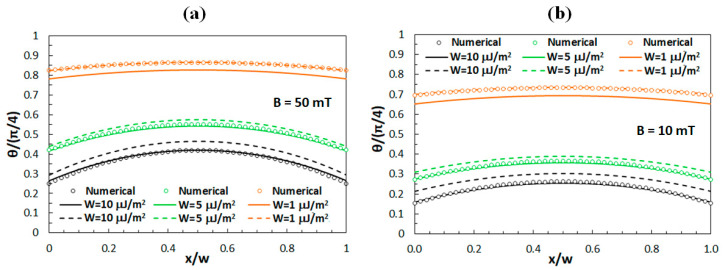
Numerical and analytical results for *θ*(*x*) for external magnetic fields of (**a**) 50 mT and (**b**) 10 mT. Circles correspond to the numerical solution, solid lines to Equation (9), and dashed lines to Equation (10). The results are given for three different values of the anchoring strength *W*.

**Figure 3 polymers-12-02355-f003:**
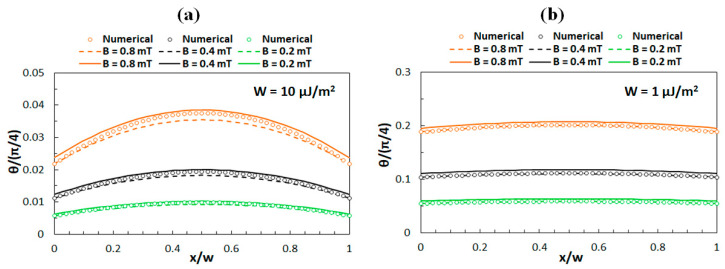
Numerical and analytical results for *θ*(*x*) for low external magnetic fields in case of (**a**) strong anchoring (*W* = 10 μJ/m^2^) and (**b**) weak anchoring (*W* = 1 μJ/m^2^). Circles correspond to the numerical solution, solid lines to Equation (9), and dashed lines to Equation (13). The results are given for three different field magnitudes.

**Figure 4 polymers-12-02355-f004:**
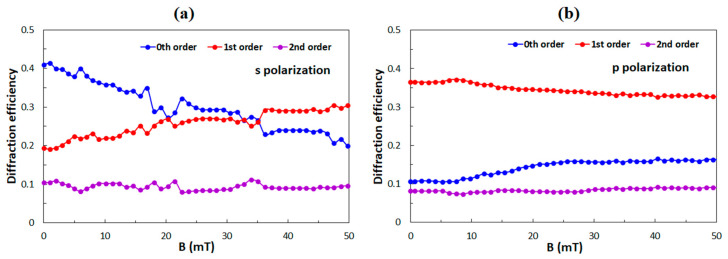
Experimental data of diffraction efficiencies of the 0th, 1st, and 2nd diffraction orders as a function of the applied external magnetic field for (**a**) the s-polarized and (**b**) for the p-polarized incident beam.

**Figure 5 polymers-12-02355-f005:**
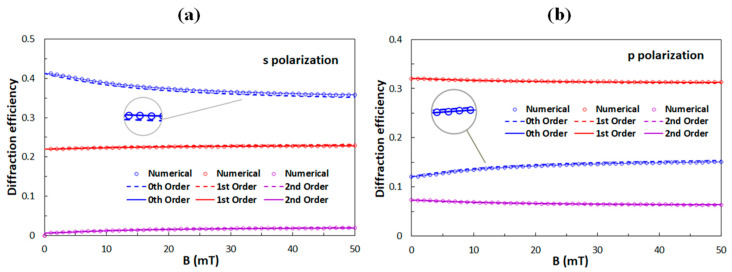
Diffraction efficiencies of the 0th, 1st, and 2nd diffraction orders as a function of the applied external magnetic field for (**a**) the s-polarized and (**b**) for the p-polarized incident beam. The anchoring strength is *W* = 5 µJ/m^2^. Circles are obtained with the numerical solution for *θ*(*x*), while solid lines are results for *θ*(*x*) obtained by Equation (9) and dashed lines for *θ*(*x*) obtained by Equation (10).

**Figure 6 polymers-12-02355-f006:**
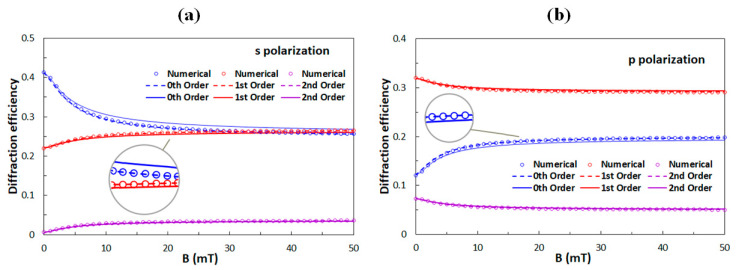
Diffraction efficiencies of the 0th, 1st, and 2nd diffraction orders as a function of the applied external magnetic field for (**a**) the s-polarized and (**b**) for the p-polarized incident beam. The anchoring strength is *W* = 1 μJ/m^2^. Circles are obtained with the numerical solution for *θ*(*x*), while solid lines are results for *θ*(*x*) obtained by Equation (9) and dashed lines for *θ*(*x*) obtained by Equation (10).

**Figure 7 polymers-12-02355-f007:**
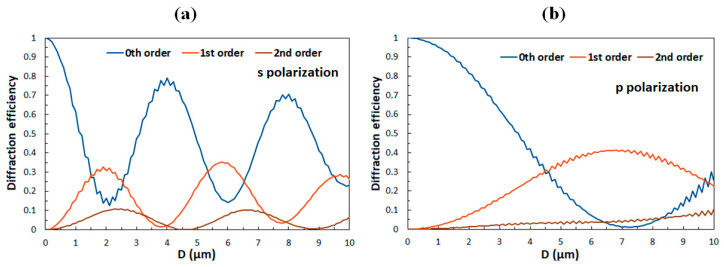
Diffraction efficiencies of the 0th, 1st, and 2nd diffraction orders as a function of the thickness *D* of the grating assembly for (**a**) the s-polarized and (**b**) the p-polarized incident beam in the zero magnetic field.

**Figure 8 polymers-12-02355-f008:**
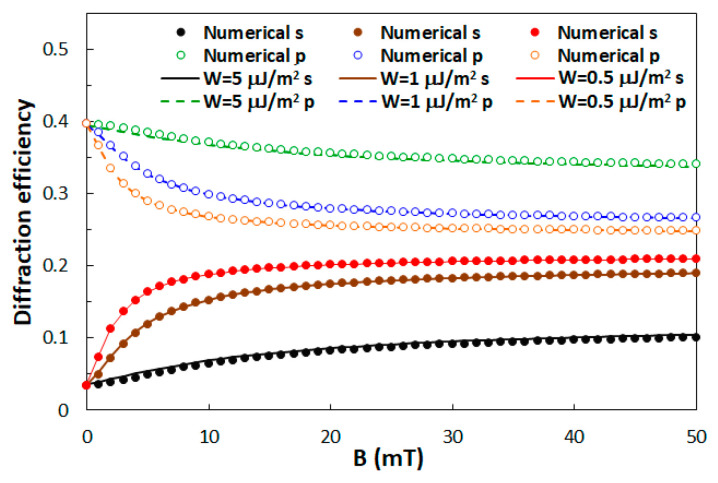
The diffraction efficiency of the 1st diffraction order for s- and for p-polarized incident beam as a function of the applied external magnetic field calculated for three different anchoring strengths.

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
