# Peer review of "Magnetically Tunable Liquid Crystal-Based Optical Diffraction Gratings"

_polymers, 2020, doi:10.3390/polym12102355_

Round 1
Reviewer 1 Report
Dear Editor,
The manuscript “Magnetically tunable liquid crystal-based optical diffraction gratings” by D. Bošnjaković, N. Sebastian and I. Drevenšek-Olenik studies theoretically diffraction of optical radiation by a periodic lattice made of polymer layers intercalated with liquid crystal (LC) doped with magnetic nanoparticles. According to the authors magnetic nanoparticles a coupled to LC molecules and magnetization of the nanoparticles is coupled to director of the LC molecules. Therefore, applying magnetic field one can control orientation of the LC director and at the end of the day optical properties of the whole diffraction grating. Authors show that diffraction efficiency may change a few tens of percent in magnetic field of order of 10 mT. The manuscript is clearly written and well prepared. Topic is relevant for the Materials journal. The presented theoretical results describe previous experiments of the same authors and can be useful for people working in the field of LC. Therefore, I recommend the manuscript for publication. However, prior to publication I want authors to make some changes in the text and make comments on my questions and concerns below.
Comments and questions
- That would be useful for reader to discuss the applicability of the energy density (1). What kind of magnetic nanoparticles are considered? Do they have their own magnetic moment or they can be paramagnetic? Can they be spherical of elongated? How are these particles attached to LC molecules? What should be particle size? Do these particles mechanically coupled to LC molecules (or via magnetic field created by the particles)?
- When the system is fabricated magnetic nanoparticles have magnetic moments that probably are randomly oriented in space and average magnetic moment is zero. When one applies magnetic field some of the particles should be co-directed with the field and some of them are counter-directed. Do both these type rotate similarly? Can the particles counter-directed to magnetic field rotate by 180 degree when the field is applied? What will happened with the director in this case?
- In their previous papers [37,38] authors assumed that director is homogeneous in space. This allowed them to explain qualitatively variation of the diffraction efficiency with magnetic field. At that the authors did not find orientation of the director as a function of magnetic field strength. Therefore, dependencies of the diffraction efficiency on magnetic field were not calculated and compared to experimental data.
What if authors solve Eq. (1) of current manuscript assuming homogeneity of the director. In this case, solutions can be found easily without any numerical simulations. Would this approach allow describing experimental data? Would it be much worse that the results obtained assuming inhomogeneity of the director?
- While author have experimental data from their previous work, they do not directly compare results of their numerical simulations to the experimental ones. That would be useful to directly fit experimental data and simulations and plot them at the same figure.
Reviewer 2 Report
In the manuscript entitled “Magnetically tunable liquid crystal-based optical diffraction gratings” by D. Bosnjakovic et al., the optical diffraction properties of magnetically tunable ferromagnetic liquid crystal gratings are investigated. Numerical calculations are performed and analytical approximations are given in the one-elastic constant approximation for low and high magnetic fields. The manuscript is in general rather well-written and organized, and it fits in the scope of the journal Polymers. Appropriate figures are shown to support the text. However, a fairly limited amount of new insights are provided, no significant breakthroughs are reported and the manuscript can be considered to be rather incremental. Moreover, some in-depth discussion is missing, the “take-home message” is not very clear and in my opinion a (more detailed) comparison to experimental results is missing in this paper. Therefore I would not recommend the manuscript for publication in Polymers in its current form. I believe that the manuscript could be reconsidered for publication after addressing comments of the reviewers and making major revisions.
Some of my concerns are discussed below in more detail.
- A more detailed comparison to experimental results should be added in the manuscript, to support the results from the analytical model. The experimental validation is now very weak.
- The added value of the currently presented model w.r.t. the previous simple model (with the assumption of x-independence/homogeneous rotation) is not clear from the manuscript. It should be demonstrated that the current model gives a better correspondence with experimental results.
- Section 1 p.2: “the associated need of metallic electrical contacts can sometimes be a disadvantage, for instance in medical applications”. Clarify this statement: what is e.g. the disadvantage of an ITO coating?
- Section 2. Grating assembly: a more detailed description would be useful. It should be explicitly mentioned whether or not the top and bottom substrate are treated with an alignment layer (and if yes, which one). Comment on the introduction of the LC material in the polymeric ribbons: mention that the polymer ribbons are produced beforehand (?) and that the LC material is a LC + magnetic nanoplatelet mixture.
- Section 3.1. assumptions at the top of page 4: motivate the assumptions.
- Section 3.1. equation 2, 4, 11: mention that some constant terms are omitted in these equations
- Section 3.1. equation 8: provide an “intuitive explanation” for this boundary condition
- Section 3.2. figure 3: it is unclear why the numerical results in Fig. 3 (b) are not compared to equation (10) and (13), instead of (9) and (13) (weak anchoring instead of strong anchoring)? Also the explanation below figure 3 should be adjusted accordingly. A more detailed explanation is also required here: why is the comparison in Fig. 3 (a) not so good, why is it better in 3 (b)? And how can it be better in 3 (b) when this weak anchoring case is compared to the analytical approximation for strong anchoring? Something seems wrong.
- Section 4.1., figure 4: To make it consistent with the values reported in section 3, it would be better to use W=10uJ/m² in figure 4 for the strong anchoring. Moreover, it would be good to include detailed insets also for the 1st and 2nd diffraction order (as is the case for the 0th order). Now it seems that eq. (9) and (10) give rise to almost no difference in 1st and 2nd order diffraction, but some difference in 0th order diffraction, which is not consistent.
- Section 4.1. “This is because for strong anchoring there can exist a large difference between the directions of n and M,…”: it would be good to give some quantitative results here. Now only results for theta(x) are given in the manuscript and nothing about psi(x) is mentioned (for some appropriately chosen values of A, B).
- Section 4.1. A very important remark that should be addressed properly by the authors is related to the final paragraph of section 4.1. The authors mention that the impact of the nanoplatelets on the anchoring strength at the walls is very strong, even though the percentage nanoplatelets in the LC mixture is very low. This should be explained and one should comment whether the presented theoretical analysis is still valid under the assumption of adhesion of nanoplatelets at the walls. If a (relatively) large amount of nanoplatelets aggregates to the walls, this seems not to be the case (even equation (1) is not valid anymore). A detailed discussion is required here.
- Section 5. Discussion and conclusion: the take-home message should be clarified here. New insights should be stressed.
- Section 5. The authors mention that out-of-plane reorientation can happen when the interaction with the top and bottom glass substrate is taken into account, but additionally also the in-plane reorientation will be influenced by these substrates. One should comment on the relative influence of the different effects (anchoring at the ribbons, anchoring at the substrates, magnetic fields), taking into account different parameters (anchoring strength at the walls and at the substrates, cell thickness, etc.).
- References could be improved: not a lot of RECENT articles are cited, apart from articles published by the authors themselves.
Author Response
Point 1: A more detailed comparison to experimental results should be added in the manuscript, to support the results from the analytical model. The experimental validation is now very weak.
Response 1: We added figure with experimental data in the manuscript together with the text related to that figure “Those values were deduced from diffraction experiments with empty gratings, i.e. before the LC material was introduced into the polymeric scaffold. Then the scaffold was filled with the ferromagnetic LC. Figure 4 shows the experimentally obtained dependencies of diffraction efficiencies of the 0th, 1st and 2nd diffraction orders on applied external magnetic fields reported in our previous work [41]”.
At the end of subchapter 4.1, We added an explanation why there is a discrepancy between the experimental results and the numerical simulations. The explanation is “The deviation of numerical calculations from the experimental results is attributed to the fact that the only parameter that was varied in our simulations was the surface anchoring strength W, while all other parameters had fixed values as found in the literature or deduced from diffraction experiments on empty gratings. However, the effective thickness of grating D might change after the filling process with the LC, because LC orientation at top and bottom surface might be partially out of the plane. Besides this, refractive index of the SU-8 polymer ribbons np may differ from that reported in the literature due to different composition of different commercial prepolymer mixtures”.
Point 2: The added value of the currently presented model w.r.t. the previous simple model (with the assumption of x-independence/homogeneous rotation) is not clear from the manuscript. It should be demonstrated that the current model gives a better correspondence with experimental results.
Response 2: We added extra figure with the experimental data, with that we get a better correspondence with the experimental results, because in the previous article [41] the s and p polarizations were replaced.
Point 3: Section 1 p.2: “the associated need of metallic electrical contacts can sometimes be a disadvantage, for instance in medical applications”. Clarify this statement: what is e.g. the disadvantage of an ITO coating?
Response 3: This statement we have clarified as follows “the associated need of metallic electrical contacts (wire connection to ITO layers) can sometimes be a disadvantage, for instance in medical applications”.
Point 4: Section 2. Grating assembly: a more detailed description would be useful. It should be explicitly mentioned whether or not the top and bottom substrate are treated with an alignment layer (and if yes, which one). Comment on the introduction of the LC material in the polymeric ribbons: mention that the polymer ribbons are produced beforehand (?) and that the LC material is a LC + magnetic nanoplatelet mixture.
Response 4: We explained the grating assembly in more detail at the beginning of the section 2. The detailed description is “As shown in our previous work, a tunable optical diffraction grating can be constructed by introducing a mixture of liquid crystalline material and ferromagnetic nanoplatelets into a periodic scaffold of parallel polymeric ribbons, where polymer ribbons are made beforehand from a negative photoresist material (SU-8 polymer materials). The periodic scaffold was fabricated by the direct laser inscription (DLW) method (Figure 1a) (Figure 1a) [43-46]. ITO-coated glass plates, that had no additional surface treatment, support the grating structure from the bottom and the top sides and the entire assembly is glued together at the edges”.
Point 6: Section 3.1. equation 2, 4, 11: mention that some constant terms are omitted in these equations.
Response 6: We added to the manuscript after equation (2) “where constant term f0 is omitted” and after equations (4) and (11) “where constant terms and terms of the Taylor series higher than O(θ2) and O(ψ2) are omitted”.
Point 7: Section 3.1. equation 8: provide an “intuitive explanation” for this boundary condition.
Response 7: For explanation of boundary condition, we extended the sentence before equation (8) in “In order to find the values of the constants C1 and C2 we applied the boundary condition on the surface where the equilibrium solution for the director n on the boundary surface must be satisfy anchoring condition, which is in our case [53]…”.
Point 8: Section 3.2. figure 3: it is unclear why the numerical results in Fig. 3 (b) are not compared to equation (10) and (13), instead of (9) and (13) (weak anchoring instead of strong anchoring)? Also the explanation below figure 3 should be adjusted accordingly. A more detailed explanation is also required here: why is the comparison in Fig. 3 (a) not so good, why is it better in 3 (b)? And how can it be better in 3 (b) when this weak anchoring case is compared to the analytical approximation for strong anchoring? Something seems wrong.
Response 8: In Figure 3, relation (10) is not used because it refers only to strong fields above a few mT. The relation (9) was used because for higher anchoring energies (above 17 μJ/m2) it gives good agreement regardless of the magnetic field. Each relation (9, 10 and 13) has its limits for the magnetic field and anchoring energy in which they valid. From Figure 3a we can see that for low magnetic fields and anchoring strength of 10 μJ/m2 we do not have good agreement, i.e., relation (9) gives good agreement for higher anchoring strength (above 17 μJ/m2). The relation (13) gives good agreement for anchoring strength up to 8 μJ/m2 and for low magnetic field, which can be seen from Figure 3b.
Point 9: Section 4.1., figure 4: To make it consistent with the values reported in section 3, it would be better to use W=10uJ/m² in figure 4 for the strong anchoring. Moreover, it would be good to include detailed insets also for the 1st and 2nd diffraction order (as is the case for the 0th order). Now it seems that eq. (9) and (10) give rise to almost no difference in 1st and 2nd order diffraction, but some difference in 0th order diffraction, which is not consistent.
Response 9: We used anchoring strength of 5 μJ/m2 because for 10 μJ/m2 there is almost no change of diffraction efficiencies of the 0th,1st and 2nd diffraction orders depending on the magnetic field. We have not included detailed insects for the 1st and 2nd diffraction order because for these orders relations (9) and (10) give almost no difference, and only adding two more inserts would reduce the visibility of the figure.
Point 10: Section 4.1. “This is because for strong anchoring there can exist a large difference between the directions of n and M,…”: it would be good to give some quantitative results here. Now only results for theta(x) are given in the manuscript and nothing about psi(x) is mentioned (for some appropriately chosen values of A, B).
Response 10: We added quantitative results “for high magnetic field (50 mT) angle y(x) ~ 35° and angle θ(x) ~ 10°”. We are given in the manuscript only results for θ(x) because we actually need it to obtain the orientation profile of the liquid crystals, which is further required for the S4 simulations and to obtain the optical diffractive properties of the associated diffraction structure.
Point 11: Section 4.1. A very important remark that should be addressed properly by the authors is related to the final paragraph of section 4.1. The authors mention that the impact of the nanoplatelets on the anchoring strength at the walls is very strong, even though the percentage nanoplatelets in the LC mixture is very low. This should be explained and one should comment whether the presented theoretical analysis is still valid under the assumption of adhesion of nanoplatelets at the walls. If a (relatively) large amount of nanoplatelets aggregates to the walls, this seems not to be the case (even equation (1) is not valid anymore). A detailed discussion is required here.
Response 11: In experiments, we noticed that by prolonged exposures to magnetic field the sample responsivity to magnetic field decreased, but the LC alignment remained. Due to this observation, we concluded that adsorption of nanoplatelets onto polymer walls reduces the anchoring strength, bit does not change the anchoring direction. Further experiments are needed to clarify this process, before additional corrections to theoretical analysis associated with it can be made.
Point 12: Section 5. Discussion and conclusion: the take-home message should be clarified here. New insights should be stressed.
Response 12: We added in section 5 “We improve the simulations with inhomogeneous orientation profile of the director n, which is determined numerically and analytically, and the diffraction efficiency was determined depending on the external magnetic field. The research was based on the anchoring energy and to see its influence as a boundary condition on the polymer-LC interface and its influence on the diffractive properties of magnetically tunable optical diffraction gratings”.
Point 13: Section 5. The authors mention that out-of-plane reorientation can happen when the interaction with the top and bottom glass substrate is taken into account, but additionally also the in-plane reorientation will be influenced by these substrates. One should comment on the relative influence of the different effects (anchoring at the ribbons, anchoring at the substrates, magnetic fields), taking into account different parameters (anchoring strength at the walls and at the substrates, cell thickness, etc.).
Response 13: We added in section 5 “In experiments top and bottom walls are composed of an ITO-coated glass plates and typical anchoring strength of LCs at ITO surface is ~ 1 μJ/m2 [62, 63]. However, in our simulations we considered that surface anchoring strength at top and bottom layers is zero. Adding finite anchoring strength at top and bottom layers leads to further improvement of our model, where we should take into account surface interaction with the ITO-glass plates on top and bottom of the assembly, which can lead to in-plane and out-of-plane LC reorientation”.
Point 14: References could be improved: not a lot of RECENT articles are cited, apart from articles published by the authors themselves.
Response 14: We added these references in the manuscript:
Tien, C.-L.; Lin, R.-J.; Kang, C.-C.; Huang, B.-Y.; Kuo, C.-T.; Huang, S.-Y. Electrically Controlled Diffraction Grating in Azo Dye-Doped Liquid Crystals. Polym. 2019, 11, 1051. [CrossRef]
- Cao, Y.; Wang, P.‐X.; D'Acierno, F.; Hamad, W.Y.; Michal, C.A.; MacLachlan M.J. Tunable Diffraction Gratings from Biosourced Lyotropic Liquid Crystals. Adv. Mater. 2020, 1907376-1-9. [CrossRef]
Naruta, T.; Akita, T.; Uchida, Y.; Lisjak, D.; Mertelj, A.; Nishiyama. N. Magnetically controllable random laser in ferromagnetic nematic liquid crystals. Optics Express 2019, 27, 24426-24433. [CrossRef]
Korostil, A.M.; Krupa, M.M. Magneto-nematic coupled dynamics in ferromagnetic nematic liquid crystals under magnetic field. Mol. Cryst. Liq. Cryst., 2020, 699, 82–89. [CrossRef]
Solodar, A.; Cerkauskaite, A.; Drevinskas, R.; Kazansky, P.G.; Abdulhalim, I. Ultrafast laser induced nanostructured ITO for liquid crystal alignment and higher transparency electrodes. Appl. Phys. Lett. 2018, 113, 081603. [CrossRef]
Choi, G.J.; Ryu, D.G.; Gwag, J.S.; Choi, Y.; Kim, T.H.; Park, M.S.; Park, I.; Lee, J.W.; Park, J.G. Anchoring strength of indium tin oxide electrode used as liquid crystal alignment layer. J. Appl. Phys. 2019, 125, 064501. [CrossRef]
Round 2
Reviewer 2 Report
The authors made some reasonable corrections to the manuscript based on the reviewers comments. I believe the manuscript has been significantly improved and now warrants publication in Polymers.